# Associations of Neutrophil-to-Lymphocyte Ratio, Platelet-to-Lymphocyte Ratio and Monocyte-to-Lymphocyte Ratio with Osteoporosis and Incident Vertebral Fracture in Postmenopausal Women with Rheumatoid Arthritis: A Single-Center Retrospective Cohort Study

**DOI:** 10.3390/medicina58070852

**Published:** 2022-06-26

**Authors:** Byung-Wook Song, A-Ran Kim, Dong-Hyuk Moon, Yun-Kyung Kim, Geun-Tae Kim, Eun-Young Ahn, Min-Wook So, Seung-Geun Lee

**Affiliations:** 1Division of Rheumatology, Department of Internal Medicine, Pusan National University Hospital, Pusan National University School of Medicine, Busan 49241, Korea; medicaldiver@naver.com (B.-W.S.); solees17@naver.com (A.-R.K.); blackalicia@naver.com (D.-H.M.); 2Biomedical Research Institute, Pusan National University Hospital, Busan 49241, Korea; 3Division of Rheumatology, Department of Internal Medicine, Kosin University College of Medicine, Busan 49267, Korea; efmsungmo@hanmail.net (Y.-K.K.); gtah@hanmail.net (G.-T.K.); 4Division of Rheumatology, Department of Internal Medicine, Pusan National University Yangsan Hospital, Yangsan 50612, Korea; ahnmolla@hanmail.net (E.-Y.A.); thalsdnrso@naver.com (M.-W.S.)

**Keywords:** rheumatoid arthritis, osteoporosis, blood cells, biomarkers

## Abstract

*Background and Objectives*: We investigated whether nutrophil-to-lymphocyte ratio (NLR), platelet-to-lymphoycte ratio (PLR), and monocyte-to-lymphocyte ratio (MLR) are associated with the presence of osteoporosis (OP) and vertebral fractures in patients with rheumatoid arthritis (RA). *Materials and Methods*: This retrospective cohort study included 413 postmenopausal patients with RA and 200 healthy controls who underwent dual-energy X-ray absorptiometry (DEXA) between January 2005 and December 2017. DEXA examination data were defined as the index date, and all laboratory values were measured within one month from the index date. OP was defined as a T-score < −2.5, and incident vertebral fractures were defined as the first occurrence of non-traumatic fractures after the index date. NLR, PLR, and MLR measures were dichotomized by a median split (low vs. high). *Results*: The median NLR, PLR, and MLR in RA patients were significantly higher than those in controls. The frequencies of OP of the lumbar spine, hip, and either site in postmenopausal patients with RA were 24.7%, 15.5%, and 32%, respectively, and were significantly higher than those in controls. After adjusting for confounding factors, a high baseline NLR was significantly associated with OP at either site (OR = 1.61, *p* = 0.041). In addition, high baseline NLR (OR = 2.11, *p* = 0.025) and PLR (OR = 2.3, *p* = 0.011) were related with the presence OP at hip. During the follow-up period, 53 (12.8%) patients with RA developed vertebral fractures incidentally. In multivariable Cox regression models, a high baseline NLR (HR = 4.72, *p* < 0.001), PLR (HR = 1.96, *p* = 0.024), and MLR (HR = 2.64, *p* = 0.002) were independently associated with a higher risk of incidental vertebral fractures. *Conclusions*: Our data suggest that NLR, PLR, and MLR can be used as potential markers of systemic bone loss among individuals with RA.

## 1. Introduction

Rheumatoid arthritis (RA) is a chronic inflammatory arthritis of unknown etiology that results in progressive joint damage, functional disability, and various extra-articular manifestations and comorbidities [1]. In addition to juxta-articular osteopenia, generalized systemic bone loss and subsequent fragility fractures are prominent clinical features of RA which impose significant morbidity and mortality [2,3]. Previous studies have reported double the risk of osteoporosis (OP) and osteoporotic fractures in RA patients compared with the general population [2,4,5]. Traditional risk factors such as aging and menopause, as well as disease-associated factors including increased disease activity, glucocorticoid (GCs) use, and immobility, are known to contribute to the development of systemic bone loss in RA [4,5,6,7]. Despite dramatic advances in the control of disease activity [8] and available effective therapeutic options for OP [9], the incidence of fragility fractures in patients with RA has gradually increased [10]. Thus, appropriate preventive measures, including screening for high-risk RA patients with OP, are crucial for optimizing the clinical outcomes.

The neutrophil-to-lymphocyte ratio (NLR), platelet-to-lymphocyte ratio (PLR), and monocyte-to-lymphocyte ratio (MLR) have recently been recognized as novel indicators reflecting the severity of systemic inflammation in various conditions. These markers are simple, easily measurable, and inexpensive; accumulating evidence suggests their potential role in the prediction of clinical outcomes in inflammatory and immune diseases, including rheumatic disorders [11,12] malignancies [13], cardiovascular diseases [14] and cholangitis [15]. A reciprocal relationship between bone and the immune system has been noted over the past two decades, and the inflammatory response can play an important role in the development of OP [16]. Although several studies have reported the association of NLR, PLR, and MLR with postmenopausal OP [17,18,19,20,21,22,23,24,25], their role in the assessment of the risk of OP and osteoporotic fractures in patients with RA has not yet been determined. In particular, NLR, PLR and MLR were reported to be related with disease activity in RA [26,27] which is an important risk factor for systemic bone loss of RA. Hence, in the present study, we aimed to investigate whether NLR, PLR, and MLR were associated with the presence of OP and vertebral fractures in patients with RA.

## 2. Materials and Methods

### 2.1. Study Design and Subjects

This was a retrospective cohort study in which data were obtained from a tertiary university hospital in Korea. We investigated 413 consecutive postmenopausal patients with RA and 200 age-matched (±2 years) postmenopausal healthy controls who underwent dual-energy X-ray absorptiometry (DEXA) between January 2005 and December 2017. Postmenopausal status was defined as no menstruation for more than one year. All patients with RA were diagnosed by experienced rheumatologists at our hospital according to the 1987 American College of Rheumatology (formerly American Rheumatism Association) revised classification criteria for RA [28] or the 2010 American College of Rheumatology/European League Against Rheumatism classification criteria [29]. The following patients with RA were excluded from the analysis: (1) those with rheumatic diseases other than RA except for Sjogren’s syndrome; (2) those taking drugs for OP treatment including bisphosphonates, selective estrogen receptor modulators, denosumab, or teriparatide, except for calcium and/or vitamin D; (3) those with a history of spine and/or hip surgery or with metal implants in situ, which could affect the result of DEXA examination; (4) those with a history of osteoporotic fractures at the spine or hip; (5) those with concomitant hematologic disorders, malignancies, or active infection; and (6) those with an estimated glomerular filtration rate (eGFR) less than 30 mL/min/1.73 m^2^. Postmenopausal healthy controls who had undergone comprehensive routine health checkups at the health promotion center of the same hospital were randomly selected for this study. The control group had no history of rheumatic diseases, including RA, hematologic diseases, malignancies, active infection, or fragility fracture, and did not take anti-osteoporotic medications except for calcium and/or vitamin D, all of which could affect NLR, PLR, MLR, or bone mineral density (BMD). The Research and Ethical Review Board of Pusan National University Hospital approved the current study and waived the requirement for informed consent because of its retrospective design (IRB No. 2102-009-099).

### 2.2. Clinical and Laboratory Data

Data on the following variables were collected in both patients with RA and healthy controls: age, body mass index (BMI), BMD measured by DEXA, and laboratory markers including complete blood counts, C-reactive protein (CRP), and serum creatinine. BMI was calculated as weight (kg) divided by height (m^2^). NLR, PLR, and MLR were calculated by dividing the absolute neutrophil count by the absolute lymphocyte count, absolute platelet count by the absolute lymphocyte count, and absolute monocyte count by the absolute lymphocyte count. The eGFR was calculated using the formula established by the Modification of Diet in Renal Disease formula: 186 × (serum creatinine)^−1.154^ × (age)^−0.203^ × 0.742 (if female) [30]. DEXA examination data were defined as the index date, and all laboratory values were measured within one month of the index date.

The following data were obtained in patients with RA: disease duration, erythrocyte sedimentation rate (ESR), disease activity score assessed using the 28-joint count for swelling and tenderness (DAS28)-ESR, rheumatoid factor (RF), anti-cyclic citrullinated protein (CCP) antibody and medications such as disease modifying anti-rheumatic drugs (DMARDs) glucocorticoids (GCs), and calcium and vitamin D concentrations. Disease duration was defined as the time interval between the date of diagnosis of RA and the date of DEXA examination (index date). DAS28-ESR was determined as follows: = [0.56×√(tender joint count 28)]+[0.28×√(swolle joint count 28)]+[0.70×lnESR]+[0.0014×visual analog scale score ] [31]. The titer of RF was measured using a particle-enhanced immunoturbidimetric assay (range, 0–14 IU/mL), and the titer of anti-CCP antibody was assessed using a chemiluminescent microparticle immunoassay (range, 0–5 U/mL).

### 2.3. Assessment of BMD and Osteoporotic Fracture

The BMD of all study subjects was measured at the total lumbar spine (L1-4) and the left hip (femoral neck and total hip) using DEXA equipment (GE-Lunar Prodigy, GE, Madison, MA, USA). All BMD measurements were performed using the same machine but by different experienced operators according to the standardized positioning and scanning protocols. BMD was expressed as absolute values (g/cm^2^) and the standard deviation (SD), measured from the mean T-score of healthy young Korean female (T-score). The reference BMD values of Korean women were provided by the equipment manufacturers. OP in postmenopausal women was defined as a T-score at either measured site according to the World Health Organization (WHO) criteria [32].

Spine radiography was performed to evaluate osteoporotic fractures at least once a year in all patients with RA, and additional radiography was performed as needed. A vertebral fracture was diagnosed if the height of the anterior, middle, and/or posterior vertebral body was reduced by 20% compared to that of the nearest uncompressed vertebral body [33], and radiographs were interpreted by an experienced rheumatologist in a blinded fashion. An incident vertebral fracture was defined as the first occurrence of a non-traumatic fracture after the index date. In addition, the use of osteoporotic medications, including bisphosphonates, denosumab, teriparatide, and selective estrogen receptor modulators, after the index date was assessed.

### 2.4. Statistical Methods

Data are expressed as mean ± SD or (with interquartile range [IQR]) for continuous variables and number of cases with percentages for categorical variables, as appropriate. The normality of the continuous variables was tested using the Kolmogorov–Smirnov test. For group comparisons, we used the Student’s *t*-test or Mann–Whitney U test for continuous variables and the chi-square test or Fisher’s exact test for categorical variables, as appropriate. Spearman correlation coefficients (ρ) were calculated to establish correlations between continuous variables. To establish the association of baseline NLR, PLR, and MLR with OP and incident vertebral fracture in patients with RA, we developed a sequential series of logistic regression models and Cox proportional hazard regression models. Variables with *p* < 0.01 in univariable models and with clinical relevance for OP and incident vertebral fracture were included following logistic and Cox regression models. In the logistic regression models, the following variables were adjusted: Model 1 used the crude model; Model 2 adjusted for age and BMI; Model 3 further adjusted for DAS28-ESR, cumulative GCs dose, and calcium/vitamin D use; and Model 4 further adjusted for RF positivity, disease duration, and DMARDs use. The following Cox proportional hazard regression models were fitted: Model 1, unadjusted; Model 2 adjusted for age and BMI; Model 3 additionally adjusted for DAS28-ESR, cumulative GCs dose, calcium/vitamin D use, and OP medication use after the index date; Model 4 additionally adjusted for disease duration and DMARDs; and Model 5 further adjusted for femur neck BMD. NLR, PLR, and MLR were dichotomized by median split (low vs. high), which were entered into logistic regression models and Cox proportional hazard regression models as independent variables. In addition, the cumulative probability of vertebral fracture-free survival was described using the Kaplan–Meier method and was compared using the log-rank test. A two-sided *p* < 0.05 was considered to indicate a statistically significant difference, and all statistical analyses were performed using STATA version 15.0 of Windows software (StataCorp LP, College Station, TX, USA).

## 3. Results

Comparisons of the clinical and laboratory data between patients with RA and healthy controls are summarized in Table 1. The median NLR, PLR, and MLR levels in RA patients were significantly higher than those in controls. The frequencies of OP at the lumbar spine, hip, and either site in postmenopausal patients with RA were 24.7%, 15.5%, and 32%, respectively, and were significantly higher than those in controls. In addition, patients with RA had a significantly reduced BMD at all sites compared to healthy subjects. Table 2 shows the baseline disease characteristics of the patients with RA. The mean disease duration and the mean DAS28-ESR were 2.67 years and 3.12, respectively. The frequency and RF and anti-CCP antibody positivity rates were 80.1% and 77.9%, respectively. Methotrexate was the most frequently prescribed DMARD (59.3%), followed by hydroxychloroquine (41.9%) and leflunomide (18.2%), and the majority of patients with RA (82.6%) were receiving GCs.

The correlations of NLR, PLR, and MLR with clinical variables in patients with RA are shown in Table 3. NLR was significantly negatively correlated with lumbar spine BMD (ρ = −0.14, *p* = 0.005), T-score (ρ = −0.134, *p* = 0.006), femoral neck BMD (ρ = −0.247, *p* < 0.001), T-score (ρ = −0.229, *p* < 0.001), total hip BMD (ρ = −0.227, *p* < 0.001), and T-score (ρ = −0.215, *p* < 0.001). PLR was also negatively correlated with the femoral neck BMD (ρ = −0.103, *p* = 0.037), total hip BMD (ρ = −0.113, *p* = 0.021), and T-score (ρ = −0.111, *p* = 0.025). However, MLR did not show a significant correlation with BMD or T-scores. In addition, DAS28-ESR was positively correlated with NLR (ρ = 0.247, *p* < 0.001) and PLR (ρ = 0.173, *p* < 0.001), but not MLR. Notably, NLR (ρ = −0.132, *p* = 0.007), PLR (ρ = −0.138, *p* = 0.005), and MLR (ρ = −0.099, *p* = 0.044) were negatively correlated with BMI.

The results of the logistic regression models investigating the association between the NLR and the presence of OP in patients with RA are described in Table 4. In Model 1, without adjustment, a high baseline NLR was associated with an increased risk of OP at all skeletal sites. In Models 2 and 3, the association of high baseline NLR with OP at either site and OP at the hip was still statistically significant. In the full-adjusted model (Model 4), OR (95% CI) of OP at either site and OP at hip with the high baseline NLR was 4.04 (1.99–8.21) and 1.76 (0.98–3.18), respectively. Otherwise, there was no statistically significant association between high baseline NLR and lumbar spine OP after adjustment for potential confounding factors in Models 2, 3, and 4. In the non-adjusted model, a high baseline PLR and MLR were significantly associated with a higher risk of hip OP (Appendix A). Although the high baseline MLR did not remain statistically significant after adjustment for confounding factors, the high baseline PLR was related to the increased risk of hip OP in Models 2, 3, and 4. As shown in Appendix A, OR (95% CI) of hip OP in RA patients with the high baseline PLR was 2.3 (1.21–4.36) in the full-adjusted model (Model 4). Furthermore, as PLR and NLR did not show significant associations with OP at either site or OP at the lumbar spine in the non-adjusted model (data not shown), further analysis by multivariable logistic regression models was not performed.

During the median follow-up period of 61 months, 53 (12.8%) postmenopausal patients with RA developed incidental vertebral fractures. The one-year and five-year fracture-free survival rates of patients with RA were 94.8% (95% CI = 92.1–96.6%) and 88.6% (95% CI = 84.6–91.5%), respectively (Appendix A). Comparisons of fracture-free survival according to NLR, PLR, and MLR levels are plotted in Figure 1. RA patients with high baseline NLR, PLR, and MLR showed significantly worse fracture-free survival rates than those without these features (*p* < 0.001 for high vs. low baseline NLR, *p* = 0.015 for high vs. low baseline PLR, and *p* = 0.001 for high vs. low baseline MLR).

The results of the Cox proportional hazards regression models are shown in Table 5. High baseline NLR, PLR, and MLR were significantly associated with a higher risk of incidental vertebral fracture in the non-adjusted model. In the fully adjusted model (Model 5), there was a significant association between high baseline NLR (HR = 4.72, 95% CI = 2.27–9.83), PLR (HR = 1.96, 95% CI = 1.09–3.53), and MLR (HR = 2.64, 95% CI = 1.43–4.89) with an increased risk of incidental vertebral fracture.

## 4. Discussion

In this retrospective cohort study, we present a detailed analysis of the association between NLR, PLR, and MLR and the risk of OP and incident vertebral fractures in postmenopausal patients with RA. An increased NLR was found to be a significant risk factor for OP at either site and at the hip, but not at the lumbar spine; however, a higher PLR was significantly associated with hip OP only. In addition, a high baseline NLR, PLR, and MLR showed a significant independent association with an increased risk of incident vertebral fracture in patients with RA. Thus, our data provide new information on NLR, PLR, and MLR as potential biomarkers of systemic bone loss in RA.

A growing body of evidence suggests that the inflammatory processes can contribute significantly to the development of OP [34]. In epidemiologic studies, chronic inflammatory diseases, such as RA, are at risk of OP and fragility fractures [35], which supports this notion. Innate and adaptive immune cells, such as neutrophils, platelets, monocytes, lymphocytes, and pro-inflammatory mediators, play a critical role in the pathogenesis of OP [34]. Thus, inflammatory markers may reflect systemic bone loss. In the literature, CRP, a representative acute phase reactant, has been demonstrated to be inversely and independently associated with BMD [36]. Increased NLR [18,21,24,25], PLR [17,23] and MLR [19] were associated with lower BMD and a higher risk of postmenopausal OP in previous studies, and these hematological markers were recognized as inflammatory indicators, as mentioned above. In line with these observations, we found that a higher NLR was associated with an increased risk of OP in female RA patients. Moreover, increased NLR, PLR, and MLR were significantly associated with the occurrence of incident vertebral fractures in female RA patients. Of note, the association of these markers with OP and fragility fractures in RA in our data was independent of disease activity and DMARDs use. Thus, we believe that NLR, PLR, and MLR may serve as surrogate markers indicating systemic bone loss in female patients with RA, irrespective of the nature of these hematologic markers to reflect inflammation.

Although the exact mechanism by which increased NLR, PLR, and MLR are associated with OP and osteoporotic vertebral fracture in patients with RA is unclear, our results suggest that neutrophils, monocytes, platelets, and lymphocytes may be actively involved in bone metabolism in RA. In RA, neutrophils can interact with resident fibroblast-like synoviocytes in the synovium to promote the inflammatory and antigen-presenting phenotype and release neutrophil extracellular traps, leading to the formation of anti-citrullinated protein antibodies, which are key factors in the development of RA [37]. Activated neutrophils in RA patients express receptor activator of nuclear factor-κB ligand (RANKL), which contributes to osteoclast activation [34]. Monocyte/macrophages play a central role in synovial inflammation in RA by stimulating neovascularization, promoting fibroblast proliferation, releasing proinflammatory cytokines, such as tumor necrosis factor-alpha [38], and promotes bone resorption through osteoclastogenesis [34]. Activated platelets can release inflammatory prostaglandins, such as prostaglandin E2, which can perpetuate synovitis in RA [39] and stimulate RANKL expression, leading to osteoclastogenesis [40]. Neutrophils, monocytes, and platelets can cause bone loss by stimulating bone resorption. Otherwise, the action of B and T lymphocytes in bone remodeling seems intricate. Under physiological conditions, B lymphocytes, regulated by the costimulatory action of T lymphocytes, can maintain homeostatic bone turnover; however, under inflammatory conditions, activated B and T lymphocytes can promote osteoclast activation [41]. We presumed that this pathophysiological background may explain the association of NLR, PLR, and MLR with systemic bone loss in RA.

Our study had several potential limitations. First, because this was a retrospective study, the causal relationship between hematologic markers, such as NLR, PLR, MLR, and OP, in RA may not be clearly elucidated. Further prospective studies are required to confirm our results. Second, we could not obtain information that could affect OP and vertebral fractures, such as smoking and alcohol consumption, due to the retrospective nature of our study. In addition, we could not fully adjust for the effect of medications on NLR, PLR, and MLR levels. Third, we could not analyze the association between these hematological markers and hip fracture because the incidence of hip fracture during the study period was too small. Fourth, this study only evaluated female patients with RA, and additional research is warranted to confirm our results in male patients. Finally, only the NLR, PLR, and MLR at the index date were investigated in the present study. Serial changes in these markers may provide more information regarding their predictive role in OP detection in female patients with RA.

## 5. Conclusions

In summary, the present study identified that a higher NLR was associated with the presence of OP, and increased baseline NLR, PLR, and MLR had a higher likelihood of incident vertebral fracture in postmenopausal patients with RA. As vertebral fracture is a serious and unfavorable complication of systemic bone loss in RA, early and accurate screening of OP in patients with RA is important. In this light, our findings suggest that NLR, PLR, and MLR may be used as biomarkers to discriminate patients with RA at a higher risk of systemic bone loss. Because NLR, PLR, and MLR are inexpensive and easily measurable, we believe that these hematological markers may play a role in the monitoring of OP in patients with RA in real clinical practice. However, owing to the potential limitations of the present study, further prospective studies are needed to confirm our results.

## Figures and Tables

**Figure 1 medicina-58-00852-f001:**
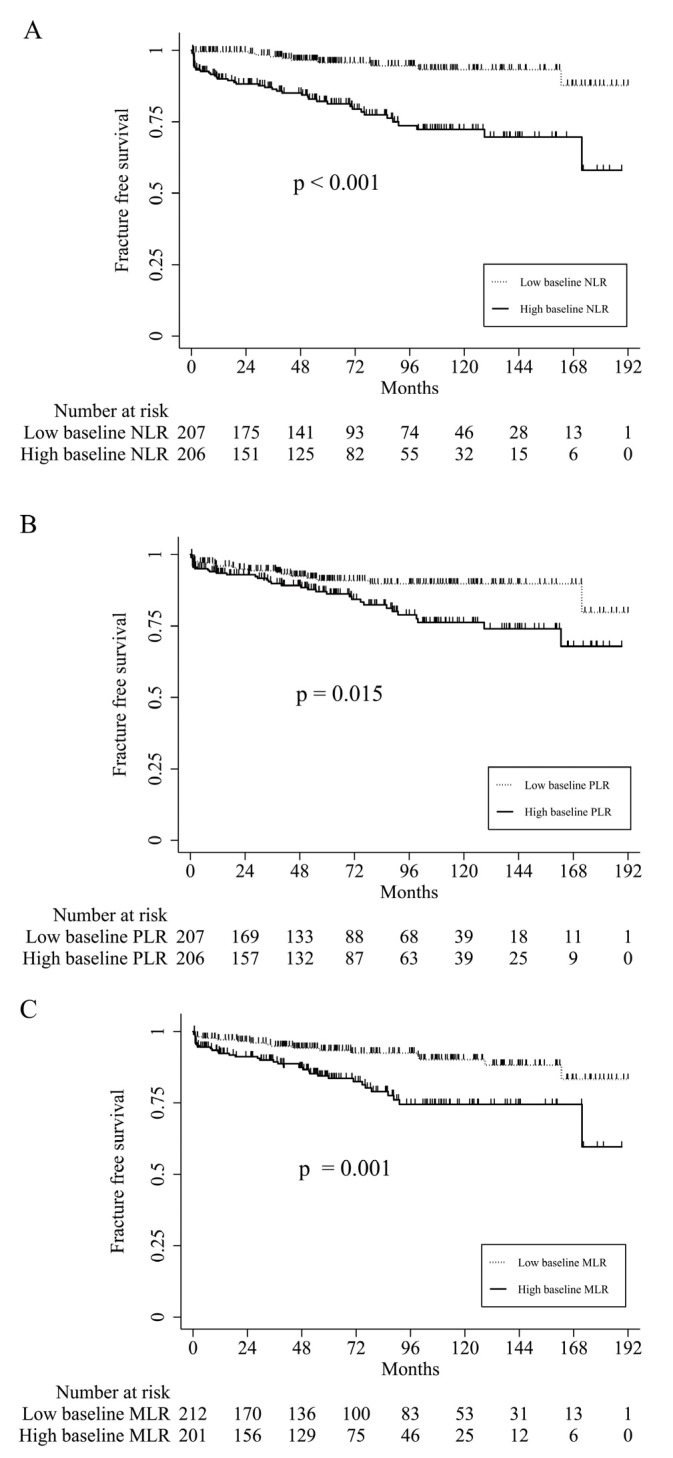
Comparisons of the fracture-free survival of patients with RA according to the neutrophil-to-lymphocyte (**A**), platelet-to-lymphocyte (**B**), and monocyte-to-lymphocyte ratios (**C**).

**Table 1 medicina-58-00852-t001:** Comparisons of clinical and laboratory data between postmenopausal women with rheumatoid arthritis and healthy controls.

	RA Patients (*n* = 413)	Controls (*n* = 200)	*p*-Value
Age, years, mean ± SD	61.9 ± 8.6	60.4 ± 6.9	0.197
NLR, median (IQR)	2.38 (1.57–3.39)	1.25 (0.97–1.57)	<0.001
PLR, median (IQR)	139.8 (108–186.3)	122.8 (102.1–146)	<0.001
MLR, median (IQR)	0.27 (0.19–0.37)	0.15 (0.12–0.18)	<0.001
CRP, mg/dL, median (IQR)	0.26 (0.07–0.91)	0.04 (0.02–0.08)	<0.001
Creatinine, mg/dL, mean ± SD	0.73 ± 0.15	0.72 ± 0.1	0.263
eGFR, mL/min/1.73m^2^, mean ± SD	91.1 ± 20.9	92.6 ± 14.5	0.278
BMI, kg/m^2^, mean ± SD	22.8 ± 2.8	23.1 ± 2.7	0.196
Lumbar spine BMD, g/cm^2^, mean ± SD	0.94 ± 0.16	1.11 ± 0.19	<0.001
Lumbar spine T score, mean ± SD	−1.5 ± 1.3	−0.1 ± 1.5	<0.001
Femoral neck BMD, g/cm^2^, mean ± SD	0.75 ± 0.12	0.87 ± 0.12	<0.001
Femoral neck T score, mean ± SD	−1.3 ± 1	−0.3 ± 1	<0.001
Total hip BMD, g/cm^2^, mean ± SD	0.82 ± 0.13	0.95 ± 0.13	<0.001
Total hip T score, mean ± SD	−1.1 ± 1	0 ± 1	<0.001
Osteoporosis at lumbar spine, *n* (%)	102 (24.7)	14 (7)	<0.001
Osteoporosis at hip, *n* (%)	64 (15.5)	3 (1.5)	<0.001
Osteoporosis at either site, *n* (%)	132 (32)	16 (8)	<0.001

RA rheumatoid arthritis, SD standard deviation, IQR interquartile range, NLR neutrophil-to-lympho cyte ratio, PLR platelet-to-lymphocyte ratio, MLR monocyte-to-lymphocyte ratio, CRP C-reactive protein, eGFR estimated glomerular filtration rate, BMI body mass index, BMD bone mineral density.

**Table 2 medicina-58-00852-t002:** Baseline disease characteristics in postmenopausal patients with rheumatoid arthritis.

	RA Patients (*n* = 447)
Disease duration, years, median (IQR)	2.67 (0.85–6.17)
ESR, mm/h, median (IQR)	24 (10–48.3)
DAS28-ESR, mean ± SD	3.12 ± 1.48
RF, IU/mL, median (IQR)	39 (20–141.4)
Anti-CCP antibody, U/mL, median (IQR)	71.9 (8.8–100)
RF positive, *n* (%)	331 (80.1)
Anti-CCP antibody positive, *n* (%)	232/298 (77.9)
DMARDs	
Methotrexate, *n* (%)	245 (59.3)
Sulfasalazine, *n* (%)	59 (14.3)
Hydroxychloroquine, *n* (%)	173 (41.9)
Leflunomide, *n* (%)	75 (18.2)
Tacrolimus, *n* (%)	10 (2.4)
bDMARDs, *n* (%)	17 (4.1)
GCs, *n* (%)	341 (82.6)
Cumulative GCs dose, g, median (IQR)	4.05 (0–10.76)
Calcium and/or vitamin D, *n* (%)	97 (23.5)

RA rheumatoid arthritis, IQR interquartile range, ESR erythrocyte sedimentation rate, DAS28-ESR disease activity score assessed using the 28-joint count for swelling and tenderness-ESR, SD standard deviation, RF rheumatoid factor, CCP cyclic citrullinated protein, DMARDs disease modifying anti-rheumatic drugs, bDMARDs biologic DMARDs, GCs glucocorticoids.

**Table 3 medicina-58-00852-t003:** Baseline disease characteristics in postmenopausal patients with rheumatoid arthritis.

	1	2	3	4	5	6	7	8	9	10	11	12	13	14
1. NLR	-	-	-	-	-	-	-	-	-	-	-	-		
2. PLR	0.625 **	-	-	-	-	-	-	-	-	-	-	-		
3. MLR	0.624 **	0.478 **	-	-	-	-	-	-	-	-	-	-		
4. Lumbar spine BMD, g/cm^2^	−0.14 **	−0.081	−0.043	-	-	-	-	-	-	-	-	-		
5. Lumbar spine T score	−0.134 **	−0.081	−0.049	0.987 **	-	-	-	-	-	-	-	-		
6. Femoral neck BMD, g/cm^2^	−0.247 **	−0.103 *	−0.091	0.482 **	0.487 **	-	-	-	-	-	-	-		
7. Femoral neck T score	−0.229 **	−0.096	−0.095	0.442 **	0.465 **	0.982 **	-	-	-	-	-	-		
8. Total hip BMD, g/cm^2^	−0.227 **	−0.113 *	−0.078	0.525 **	0.529 **	0.873 **	0.853 **	-	-	-	-	-		
9. Total hip T score	−0.215 **	−0.111 *	−0.088	0.182 **	0.506 **	0.86 **	0.867 **	0.985 **	-	-	-	-		
10. DAS28-ESR	0.247 **	0.173 **	0.082	−0.233 **	−0.199 **	−0.198 **	−0.144 **	−0.215 **	−0.165 **	-	-	-		
11. Age, years	0.136 **	−0.002	0.102 *	−0.302 **	−0.306 **	−0.306 **	−0.463 **	−0.433 **	−0.429 **	0.103 *	-	-		
12. BMI, kg/m^2^	−0.132 **	−0.138 **	−0.099 *	0.162 **	0.154 **	0.154 **	0.21 **	0.27 **	0.249 **	−0.101 *	−0.004	-		
13. Cumulative GCs dose	0.095	0.051	0.093	−0.048	−0.056	−0.154 *	−0.164 *	−0.174 **	−0.184 **	−0.135 *	−0.007	−0.062		
14. Current GCs use	0.093	0.094	0.092	−0.089	−0.092	−0.121*	−0.131 *	−0.128 *	−0.142 *	−0.05	0.064	−0.074	0.552 **	

* *p* < 0.05, ** *p* < 0.001. NLR neutrophil-to-lymphocyte ratio, PLR platelet-to-lymphocyte ratio, MLR monocyte-to-lymphocyte ratio, BMD bone mineral density, DAS28-ESR disease activity score assessed using the 28-joint count for swelling and tenderness-erythrocyte sedimentation rate, BMI body mass index, GCs glucocortidoids.

**Table 4 medicina-58-00852-t004:** Association between baseline neutrophil-to-lymphocyte ratio and osteoporosis in patients with rheumatoid arthritis evaluated by logistic regression models.

Dependent Variable	OP at Either Site		Hip OP		Spine OP	
Independent Variable	High Baseline NLR		High Baseline NLR		High Baseline NLR	
	OR (95% CI)	*p*-Value	OR (95% CI)	*p*-Value	OR (95% CI)	*p*-Value
Model 1	1.98 (1.3–3.02)	0.001	3.02 (1.68–5.41)	<0.001	1.61 (1.03–2.54)	0.038
Model 2	1.59 (1.01–2.5)	0.045	2.26 (1.2–4.25)	0.011	-	-
Model 3	1.59 (1.01–2.5)	0.045	2.17 (1.15–4.11)	0.017	-	-
Model 4	1.61 (1.03–2.58)	0.041	2.11 (1.1–4.2)	0.025	-	-

Model 1: no variable adjusted. Model 2: age and BMI adjusted. Model 3: age, BMI, DAS28-ESR, cumulative GCs dose and calcium/vitamin D adjusted. Model 4: age, BMI, DAS28-ESR, cumulative GCs, calcium/vitamin D, RF positivity, disease duration and DMARDs use adjusted. OP osteoporosis, NLR neutrophil-to-lymphocyte ratio, OR odds ratio, CI confidence interval, BMI body mass index, DAS28-ESR disease activity score assessed using the 28-joint count for swelling and tenderness-erythrocyte sedimentation rate, GCs glucocorticoids, RF rheumatoid factor, DMARDs disease modifying anti-rheumatic drugs.

**Table 5 medicina-58-00852-t005:** Hazard ratios of baseline neutrophil-to-lymphocyte ratio, platelet-to-lymphocyte ratio and monocyte-to-lymphocyte ration for incident vertebral fracture in patients with rheumatoid arthritis.

Dependent Variable	Incident Vertebral Fracture		Incident Vertebral Fracture		Incident Vertebral Fracture		
Independent Variable	High Baseline NLR		High Baseline PLR		High Baseline MLR		
	HR (95% CI)	*p*-Value	HR (95% CI)	*p*-Value	HR (95% CI)	*p*-Value	
Model 1	5 (2.51–9.97)	<0.001	2 (1.13–3.53)	0.017	2.61 (1.46–4.67)	0.001	
Model 2	4.46 (2.22–8.94)	<0.001	2.01 (1.14–3.55)	0.016	2.33 (1.29–4.21)	0.005	
Model 3	4.46 (2.22–8.94)	<0.001	2.01 (1.14–3.55)	0.016	2.31 (1.28–4.17)	0.006	
Model 4	4.49 (2.24–9)	<0.001	2.02 (1.14–3.57)	0.015	2.32 (1.28–4.19)	0.005	
Model 5	4.72 (2.27–9.83)	<0.001	1.96 (1.09–3.53)	0.024	2.64 (1.43–4.89)	0.002	

Model 1: no variable adjusted. Model 2: age and BMI adjusted. Model 3: age, BMI, DAS28-ESR, cumulative GCs dose, calcium/vitamin D and osteoporosis medications adjusted. Model 4: age, BMI, DAS28-ESR, cumulative GCs dose, calcium/vitamin D, osteoporosis medications, disease duration and DMARDs use adjusted. Model 5: age, BMI, DAS28-ESR, cumulative GCs dose, calcium/vitamin D, osteoporosis medications, disease duration, DMARDs use and femoral neck BMD adjusted. NLR: neutrophil-to-lymphocyte ratio, PLR: platelet-to-lymphocyte ratio, MLR: monocyte-to-lymphocyte ratio, HR: hazard ratio, CI confidence interval, BMI: body mass index, DAS28-ESR: disease activity score assessed using the 28-joint count for swelling and tenderness-erythrocyte sedimentation rate, GCs glucocorticoids, RF: rheumatoid factor, DMARDs: disease modifying anti-rheumatic drugs, BMD: bone mineral density.

## Data Availability

Data from this study are available on request to the corresponding author of the study.

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
