# Peer review of "Associations of Neutrophil-to-Lymphocyte Ratio, Platelet-to-Lymphocyte Ratio and Monocyte-to-Lymphocyte Ratio with Osteoporosis and Incident Vertebral Fracture in Postmenopausal Women with Rheumatoid Arthritis: A Single-Center Retrospective Cohort Study"

_medicina, 2022, doi:10.3390/medicina58070852_

Round 1

Reviewer 1 Report

First of all, since NLR, PLR and MLR values are more and more discussed lately as potential biomarkers for various inflammatory and oncological diseases associations, the topic of the article is quite important and could have real potential practical use. Regarding methodology and patient selection increased cortisone use in RA cohort could bias the results since glucocorticoid induced osteoporosis is a well defined entity. Also, since mean DAS28 was 3.12 (low disease activity) and nowadays RA standard of care relies especially on DMARDs and not on long term cortisone use, how can be explained that most of the patients (82.6%) were taking GCs.

About results, how can you explain that only high baseline NLR was significantly associated with OP at either sites but high baseline NLR and PLR were related with the presence OP at hip. Also, high baseline NLR, PLR and MLR were independently associated with a higher risk of incidental vertebral fractures (and not with hip fractures as suggested by NLR and PLR correlations with low hip BMD).

Terminology suggestion: since osteoporosis is defined by the finding of a value below -2.5 at either measured site, using words like „osteoporosis at the hip or osteoporosis at the lumbar spine”  it’s a little bit strange. Please consider to use „T score values below – 2.5” at mentioned sites.

Author Response

First of all, since NLR, PLR and MLR values are more and more discussed lately as potential biomarkers for various inflammatory and oncological diseases associations, the topic of the article is quite important and could have real potential practical use. Regarding methodology and patient selection increased cortisone use in RA cohort could bias the results since glucocorticoid induced osteoporosis is a well defined entity. Also, since mean DAS28 was 3.12 (low disease activity) and nowadays RA standard of care relies especially on DMARDs and not on long term cortisone use, how can be explained that most of the patients (82.6%) were taking GCs.

: Thank you for your kind comment. Although international guidelines for RA management recommend the use of GCs as low as possible, there is large variability among rheumatologists in the prescribing of GCs for patients with RA, as reported by George et al. (Variability in Glucocorticoid Prescribing for Rheumatoid Arthritis and the Influence of Provider Preference on Long-Term Use of Glucocorticoids. Arthritis Care Res (Hoboken) 2021 Nov;73(11):1597-1605). Thus, the high proportion of RA patients receiving GCs in our study can be possible in real practice. For example, a large-scale retrospective cohort study by Wallace et al which investigated pattern of GCs prescription in RA patients found that 6,771/9,221 (72.8%) RA patients were taking GCs (Patterns of glucocorticoid prescribing and provider-level variation in a commercially insured incident rheumatoid arthritis population: A retrospective cohort study. Semin Arthritis Rheum 2020 Apr;50(2):228-236).

About results, how can you explain that only high baseline NLR was significantly associated with OP at either sites but high baseline NLR and PLR were related with the presence OP at hip. Also, high baseline NLR, PLR and MLR were independently associated with a higher risk of incidental vertebral fractures (and not with hip fractures as suggested by NLR and PLR correlations with low hip BMD).

: The exact machanism by which NLR and PLR but not MLR were assoicated with the risk or OP is not clear. We conjecture that neutrophil and platelet might play a more important role in systemic bone loss in RA as compared with monocyte.

Unlike postmenopausal osteoprosis, marked bone loss in the hip but relatively preserved axial bone mass in RA pateitns have been reported in previous studies. (Increased frequency of osteoporosis and BMD below the expected range for age among South Korean women with rheumatoid arthritis. Int J Rheum Dis 2012;15:289–96. Bone mineral density andfrequency of osteoporosis in female patients with rheumatoid arthritis:results from 394 patients in the Oslo County Rheumatoid Arthritis register. Arthritis Rheum 2000;43:522–30. Reduced bone mineral density in male rheumatoid arthritis patients: frequencies and associations with demographic and disease variables in ninety-four patients in the Oslo County Rheumatoid Arthritis Register. Arthritis Rheum 2000;43:2776–84. Bone mineral density and frequency of osteoporosis among Vietnamese women with early rheumatoid arthritis. Clin Rheumatol 2011;30:1353–61).

This result suggest that bone loss of hip (cortical bone) is more vulnerable to inflammatory status than that of lumbar spine (trabecular bone). Thus, inflammatory markers such as NLR and PLR may be more closely related with hip osteoporosis compared with lumbar spine osteoporosis.

Because the case of hip fractures was small, we did not evaluate the associaton of NLR, PLR MLR with hip fractures in RA pateitns in this study. It is known that femral neck BMD is closely related to major osteoporotic fractures including vertebral fractures. In the FRAX tool, femoral neck BMD is included among the varibles for predicting major osteoporotic fracures. This notion may explain why NLR and PLR were assoicated with incident vertebral fractures although these markers are more closely associated with hip OP rather that lumbar spine OP.     

Terminology suggestion: since osteoporosis is defined by the finding of a value below -2.5 at either measured site, using words like „osteoporosis at the hip or osteoporosis at the lumbar spine”  it’s a little bit strange. Please consider to use „T score values below – 2.5” at mentioned sites.

: We corrected this sentence as you suggest.

Reviewer 2 Report

The rationale for the study should be more clearly defined- currently it is hard to understand why NLR/PLR/MLR are chosen as markers of osteoporosis. NLR/PLR seem to be markers of inflammatory activity that is an established risk factor for osteoporosis development in RA. The authors cite mainly their own previous work when describing the study rationale- explanation of the topic in the light of the current literature should be wider. Currently it is hard to understand what this approach could add to what is already known.

The proposed markers can be associated with glucocorticoid usage- correlation between GCS (both current usage and cumulative dose) and the markers should be added to table 3. 

CUmulative GCS dose is included in the logistic regression models, but not current usage that can have an effect on the marker values- to improve the quality of the manuscript I would advise this to be added.

Author Response

The rationale for the study should be more clearly defined- currently it is hard to understand why NLR/PLR/MLR are chosen as markers of osteoporosis. NLR/PLR seem to be markers of inflammatory activity that is an established risk factor for osteoporosis development in RA. The authors cite mainly their own previous work when describing the study rationale- explanation of the topic in the light of the current literature should be wider. Currently it is hard to understand what this approach could add to what is already known.

: Thank you for your kind comment. Many studies reported that NLR, PLR MLR were associated with disease activity in RA. Although the etiology of osteoporosis in RA is multifactorial, disease activity is one of the most important factors for systemic bone loss in RA. Thus, we hypothesized that NLR, PLR, MLR may be associated with osteoporosis and osteoporotic fracture in RA. We added this notion briefly in the Introduction section and cited addition references.

The proposed markers can be associated with glucocorticoid usage- correlation between GCS (both current usage and cumulative dose) and the markers should be added to table 3. 

: We added the correlation of current use and cumulative dose of GCs with clinical variables in Table 3, as you recommend. The cumulative dose of GCs had a stronger correlation with BMD and T-score of femoral neck and total hip than the current use of GCs.   

CUmulative GCS dose is included in the logistic regression models, but not current usage that can have an effect on the marker values- to improve the quality of the manuscript I would advise this to be added.

: When the current use instead of the cumulative glucocorticoid dose was entered into the logistic regression model, the results on the relationship between NLR, PLR, and MLR with osteoporosis and incident vertebral fracture were same (We added results  in the attached file).  As cumulative GCs dose  is kwon to be more closely related with  osteoporosis and osteoporotic fracture than current use of GCs, we  selected  cumulative GCs as  independent variable in the logistic and Cox regression models. 
